# Accelerating the solar-thermal energy storage via inner-light supplying with optical waveguide

Yafang Zhang[1], Jiebin Tang[2], Jialin Chen[2], Yuhai Zhang [2], Xiangxiang Chen[2], Meng Ding[1], Weijia Zhou [2], Xijin Xu[1], Hong Liu [2,3] ✉ & Guobin Xue [2] ✉

Solar-thermal storage with phase-change material (PCM) plays an important role in solar energy utilization. However, most PCMs own low thermal conductivity which restricts the thermal charging rate in bulk samples and leads to low solar-thermal conversion efficiency. Here, we propose to regulate the solar-thermal conversion interface in spatial dimension by transmitting the sunlight into the paraffin-graphene composite with side-glowing optical waveguide fiber. This inner-light-supply mode avoids the overheating surface of the PCM, accelerates the charging rate by 123% than that of the traditional surface irradiation mode and increases the solar thermal efficiency to ~94.85%. Additionally, the large-scale device with inner-light-supply mode works efficiently outdoors, indicating the potential of this heat localization strategy in practical application.

To alleviate the shortage of fossil fuel and the environmental pollution, economical and sustainable solar energy has gained tremendous attention in recent years[1–4]. Among various technologies of solar energy utilization, solar-thermal energy storage (STES) technologies are widely studied to counter the mismatch between supply and energy demand as solar energy is intermittent and weather-dependent[5–7]. The STES technology based on phase change materials (PCMs) is especially studied owing to low cost, high volumetric energy storage density, and relatively stable phase transition temperature range[8–12]. Usually, solar-to-thermal conversion and thermal transport process are involved in STES technology. The thermal conductivity of most PCMs is generally lower than $1\,W\,m^{-1}\,K^{-1}$, which severely impedes the efficient thermal transfer in bulk PCMs[13,14]. The slow movement of charging interface and low thermal energy storage rate restrict the solar-to-thermal conversion efficiency and cause potential overheating issues.

The most common strategy to accelerate the charging rate is enhancing the thermal conductivity of PCMs. For example, carbon materials like expanded graphite (EG) and carbon fiber, boron nitride (BN), metal nanoparticles, and metallic oxide with ultra-high thermal

conductivity are widely studied as thermal transfer fillers[15–24]. However, the large interfacial thermal resistance reduces the effective thermal transfer between these discrete additives and PCMs[25]. As a result, the composite PCMs with high amounts of additives just show moderate thermal transfer capability. In addition, the high loadings of the filler often lead to the reduction of thermal energy storage capacity of the PCMs as the fillers have no contribution to the phase-change fusion enthalpy[26–28]. The filler with high content is also easy to agglomerate during the solid/liquid phase transition process. In this regard, a variety of three-dimensional (3D) conducting/light-absorbing porous networks such as metal foams, carbon-based sponges, and carbonized aerogels are proposed as the supporting matrix of PCMs[29–38]. The 3D porous frameworks offer thermally conducting pathways for energy transportation and show higher thermal conductivity enhancement of PCMs than the discontinuous additives in PCMs[39–47]. Moreover, supporting material such as metal organic frameworks (MOFs) have also recently been used to encapsulate PCMs profiting from the advantages of high porosity, regular pores, and large specific surface area[48–53]. According to Beer-Lambert law, light energy attenuates exponentially in PCMs with the size of storage materials. Therefore, it still relied on

¹School of Physics and Technology, University of Jinan, Jinan 250022, China. ²Institute for Advanced Interdisciplinary Research (iAIR), School of Chemistry and Chemical Engineering, University of Jinan, Jinan 250022, China. ³State Key Laboratory of Crystal Materials, Shandong University, Jinan 250100, China. ✉e-mail: hongliu@sdu.edu.cn; ifc_xuegb@ujn.edu.cn

thermal diffusion to accomplish the charging process in above-mentioned composite PCMs and fast solar-thermal energy storage within bulk composite PCMs is still a grand challenge. Recently, Deng's group reported an enhanced photon-transport-based charging phase change system in which the dispersion of optical absorbers was tuned dynamically with an external magnetic field. This approach achieved fast charging rate and high solar-thermal energy conversion efficiency[54,55]. They also reported roll-to-roll solar-thermal energy harvesting system to shorten heat-diffusion distance[56]. In the long term, dynamic tuning charging interface provides a superior means of speeding up the charging rate and enhancing heat transfer efficiency. However, external actuator is needed in this process which enhances the complexity of the STES. Further studies to boost the heat storage rate by altering the solar-thermal conversion interface in spatial dimension are needed.

In this work, we demonstrate an inner-light-supply mode to accelerate the solar-thermal energy charging rate. As shown in Fig. 1 (right), the sunlight is concentrated by a concentrator and transmits into a side-glowing optical fiber. After multiple scattering and refracting, the rays output from the side face of the fiber and are transferred to heat by the graphene dispersed in paraffin. Comparing to traditional surface irradiation mode (Fig. 1a), this inner-light-supply strategy can greatly boost the heat storage rate as the charging interface is optimized in spatial dimension based on the optical waveguide effect. We show that the charging rate in inner-light-supply mode is about ~2.5 times of that in surface irradiation mode. The fast movement of charging interface leads to a high energy efficiency of 94.85%. At the meantime, long-distance transmission characteristic of the optical fiber enables this inner-light-supply mode STES system to work in large scale. This heat localization strategy is universally applicable for solid–solid, solid–gas, and liquid–gas STES and offers an avenue to diverse solar-thermal applications.

## Results

### Characterization of paraffin-graphene composite

We chose paraffin as the typical phase change matrix and few-layer graphene as the optical absorber. SDBS was used as the surfactant to enhance the dispersion ability of graphene[57]. As shown in Fig. 2a and Supplementary Fig. 1, the graphene is uniformly dispersed in paraffin. To quantitatively characterize the light absorption characteristic of paraffin-graphene composite, we measured the transmittance and reflectivity spectra ranging from 300 to 2500 nm. As shown in Fig. 2b and Supplementary Fig. 2, the absorption of pure paraffin is relatively weak for its high transmittance and reflectivity and can be obviously enhanced with a small amount of graphene. The reflectivity is lower than 10% when the loading is over 0.02 wt%, indicating that over 90% incident light will be absorbed by the composite. Meanwhile, the transmittance is above 10% with the loading of 0.02 wt%, meaning light can transport over 1 mm in this composite. Then phase change behavior was characterized by differential scanning calorimetry (DSC). Figure 2c shows the DSC curves of pure paraffin and paraffin-graphene composites with graphene loading ranging from 0.01 wt% to 0.06 wt%. The DSC plot shows that the onset melting temperature of pure paraffin is about 40.5 °C with a peak position at 48 °C and the onset crystallization temperature is about 47.5 °C with the peak position at 45 °C. In comparison, the onset melting temperature of all the composites shifts to about 42 °C, and the peak position increases to 51.5 °C. The onset crystallization temperature shifts to 51.5 °C and the peak position is about 48.5 °C. The increase of phase change temperature of paraffin-graphene composite attributes to the addition of graphene which could possible affects surrounding organic molecules and delays the structure change of paraffin[58]. On the basis of the enclosed area in the DSC curve, the phase change enthalpy of PCMs can be calculated (Fig. 2d). We can see that the phase change enthalpy of pure paraffin is 145 J g⁻¹ and reaches a maximum value of 151 J g⁻¹ after compositing with 0.02 wt% of graphene. When the graphene loading is higher than 0.02 wt%, the phase change enthalpy is reduced with the increase of graphene. Considering both the light absorption ability and the phase change enthalpy characteristic, we chose the composite with 0.02 wt% graphene to conduct the following studies. The thermal conductivity of this composite is about 0.3 W m⁻¹ K⁻¹, little higher than that of pure paraffin (0.26 W m⁻¹ K⁻¹). Therefore, thermal conductivity enhancement has little contribution to the observed higher charging rate in the STES system with inner-light-supply mode in this work.

### Side glowing properties of the optical fiber

In this work, we used the low-cost plastic optical fiber (POF) as an optical waveguide to transmit light into the inner of the paraffin-graphene composite (Fig. 3a). Side-glowing optical fibers are prepared based on the "similarity-intermiscibility" theory. As the mixture of the acetone and hexane agent with the volume ratio of 2:3 has approximate solubility parameter to PMMA, this mixture agent can dissolve PMMA and destroy the surface structure of the POF. As shown in SEM images, the surface of original POF is smooth (Fig. 3b, c). After being etched with the mixture agent, the surface of POF becomes rugged and many potholes appear (Fig. 3d, e). With the increase of treatment time, these potholes become bigger (Supplementary Fig. 3). When the light propagates in the etched optical fiber, total reflection condition can't be satisfied due to the present of superficial injury and the light is output at the damaged lateral surface of the optical fiber. As shown in Fig. 3f, the light intensity on the lateral surface of the original optical fiber is very low. After being etched with the mixture agent, the intensity of lateral light obviously increases (Fig. 3g). The lateral light

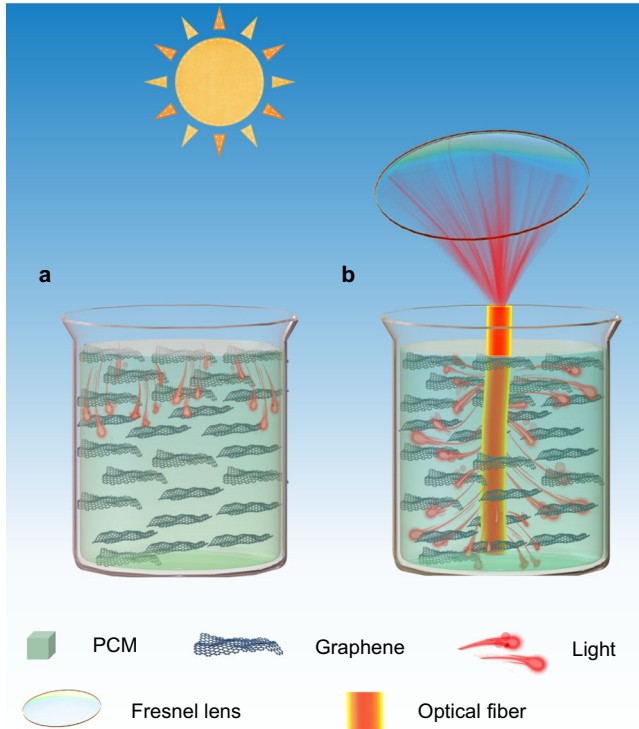

**Fig. 1 | Schematic diagram of the phase-change thermal storage system. a** In traditional surface irradiation mode, additive such as graphene is used to enhance the light absorption and thermal conductivity of the PCM. Solar-thermal conversion process occurs at the surface of the PCM. **b** To further accelerate the thermal charging rate, inner-light-supply mode is achieved with optical fiber. The sunlight is focused by collecting lens and then transmits into the PCM with the side-glowing optical fiber after multiple scattering and refracting. The solar-thermal conversion interface is localized in the inner of the PCMs, in which well-dispersed graphene converts light to heat and heat is stored in PCM accompanying phase change process.

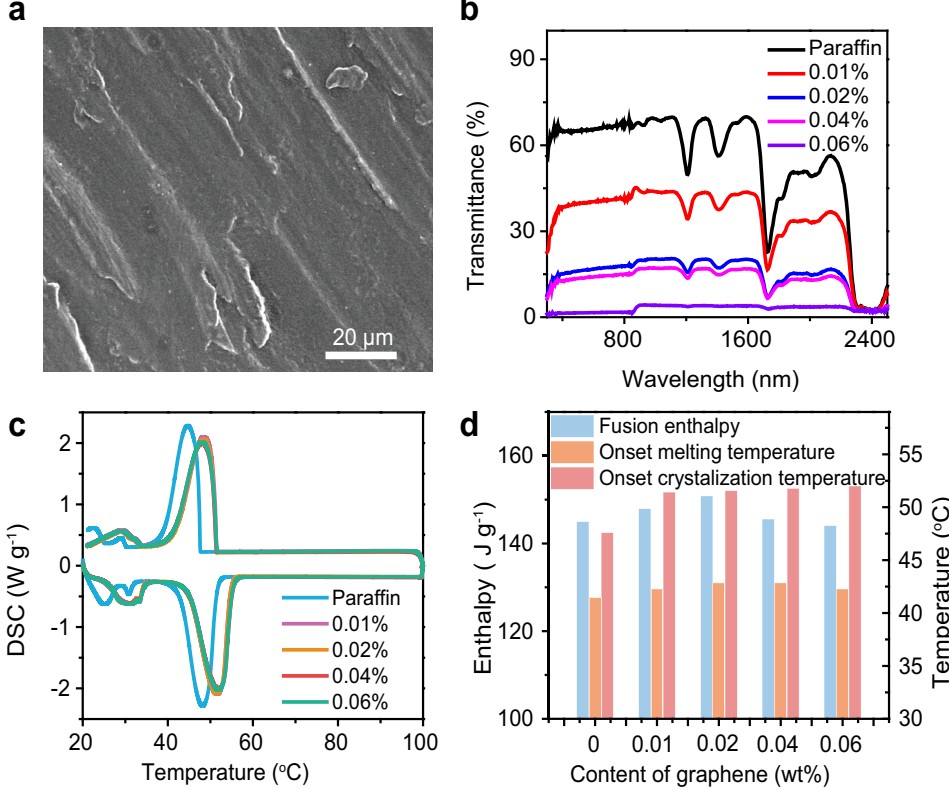

**Fig. 2 | Characterization of paraffin-graphene composite. a** SEM image of the surface morphology of paraffin-graphene composite. **b** Transmittance spectra of paraffin-graphene composites with a thickness of 1 mm in the range of 300 to 2500 nm spectra. **c** DSC curves of pure paraffin and paraffin-graphene composites with different loading of graphene. **d** Comparison of fusion phase-change enthalpy, melting and solidification temperatures of pure paraffin and paraffin-graphene composites.

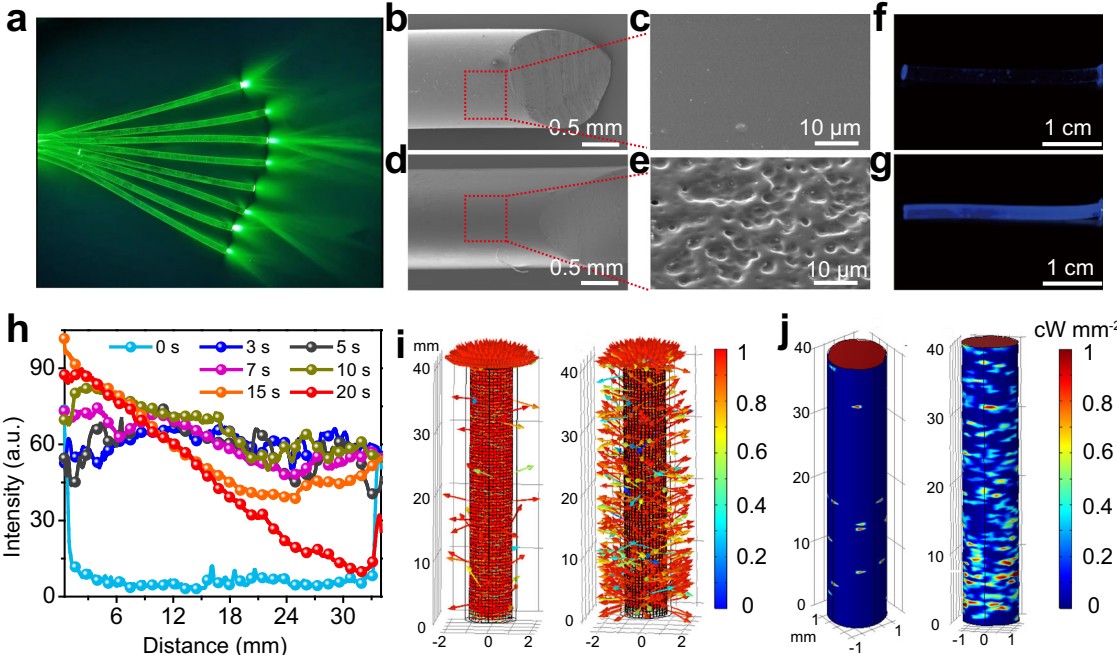

**Fig. 3 | Side glowing characterization of POF. a** Optical image of POF. **b–e** SEM images of POF treated with the organic solvent for 0 s (**b**, **c**) and 10 s (**d**, **e**). **f**, **g** Optical photographs of the side-glowing POF treated with the organic solvent for 0 s (**f**) and 10 s (**g**). **h** The light intensity distribution of POF treated with the organic solvent for different time. **i** Ray trajectories in POF with different size pothole, 5 μm (left) and 25 μm (right). The color expression for the refracted rays indicates the transmittance as they exit the POF. **j** Deposited ray power on the lateral and bottom surface of POF with different size pothole, 5 μm (left) and 25 μm (right).

intensity of optical fiber with different etched time is displayed in Fig. 3h and Supplementary Fig. 4. When the treatment time is less than 10 s, the lateral light intensity of the fiber enhances gradually from left to right and becomes equally distributed at 10th s. When the etching time is over 10 s, the uniformity of lateral light intensity begins to aggravate. Therefore, the etching time of 10 s is fixed to prepare homogeneous lateral-glowing optical fiber. It should be noted that the side glowing properties is also affected by length and diameter of the fiber. The optimized etching time is got at present length of the fiber. If the length of the fiber is various, the etching condition or etching method, such as laser corrosion, should be adjusted. As shown in Supplementary Fig. 5, commercial side-glowing optical fiber with large-area can be easily got. Thus the depth limit of uniform side-glowing fiber should be considered and may be easily solved in practical application. We then used a finite element method based on the COMSOL Multiphysics software to comprehensively explore the optical propagating process in optical fiber. We tracked the trajectories of 1000 articles of ray in optical fiber with different pothole size of 5, 25, and 45 μm. As shown in Fig. 3i and Supplementary Fig. 6a, the light path

is obviously affected by the pothole size. When the radius of these potholes is 5 μm, only a small part of the beam emits from lateral face of the fiber. Huge amount of ray emits relatively uniformly from lateral face of the fiber when the radius of the potholes increases to 25 μm. When the radius of the potholes is 45 μm, vast majority of the ray emits from lateral face of the fiber near the light incident end. The deposited ray power on the lateral and bottom side of the fiber is shown in Fig. 3j and Supplementary Fig. 6b. We can see more visually that the radius of these potholes affects the side-glowing properties. With these potholes get bigger, the side-glowing power keeps growing while the distribution of beam on lateral face of fiber is first uniform and then uneven. These theoretical simulation results are consistent with the experimental results.

### Solar-thermal storage with inner-light-supply mode

Then the side-glowing optical fiber was used in the solar-thermal storage system in laboratory conditions (Fig. 4a). The temperature evolution of the paraffin-graphene composite (0.02 wt%) under a blue laser radiation (800 mW) was recorded with an infrared (IR)

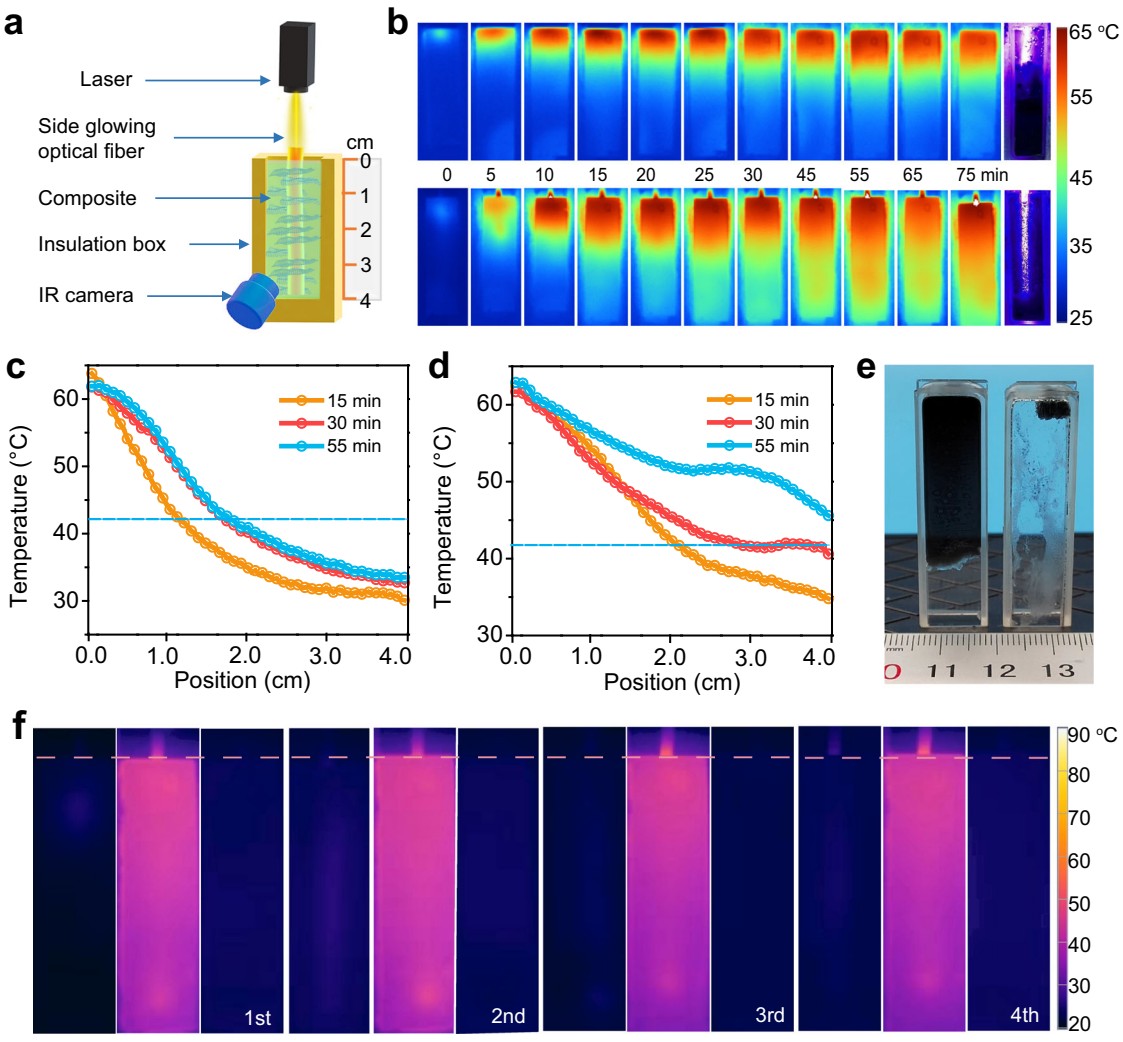

**Fig. 4 | Charging of paraffin-graphene composite. a** Schematic of experimental setup for charging of paraffin-graphene composite. A blue laser with the wavelength of 450 nm is used to illuminate onto the incident face of the POF in inner-light-supply mode and directly illuminate the surface of paraffin-graphene composite in surface irradiation mode. **b** Temperature evolution of paraffin-graphene composite in surface irradiation mode (top) and inner-light-supply mode (bottom) by prolonging the charging time from 0 min to 75 min. **c, d** Temperature distribution profiles at different charging times in surface irradiation mode (**c**) and

inner-light-supply mode (**d**). The cyan line denotes the phase change temperature of the composite (42 °C). **e** Photographs of unmelted paraffin-graphene composite in surface irradiation mode (right) and inner-light-supply mode (left) after charging for 75 min. In order to get a better view, we put the sample upside down. **f** IR thermal image of the paraffin-graphene composite in inner-light-supply mode during four phase transition cycles. During each cycle, we displayed the charging and discharging process.

camera. As shown in Fig. 4b, in surface irradiation mode only 45% of the composite transits into liquid state even the charging time extended to 75 min. In inner-light-supply mode, almost all the paraffin-graphene composite can change to liquid phase with 55 min. Then we extracted the temperature distribution from the infrared photo at the charging time of 15 min, 30 min, and 55 min (Fig. 4c, d). The horizontal axis represents the distance to the top of quartz cuvette as shown in Fig. 4a. The cyan line denoting the phase change temperature of the paraffin-graphene composite (42 °C) is used to tracked the position of the charging interface. It can be seen that about 40% of the composite transits into liquid state at 30th min in the surface irradiation mode. At the 55th min, the temperature profile has relatively small change and the solid/liquid interface advanced to 1.79 cm. By comparison, in the inner-light-supply mode, about 70% of the solid composite (2.8 cm) transits into liquid state at 30th min and the rest 30% of the composite is around the onset melting temperature. By prolonging the charging time to 55 min, all of the solid composite transits into liquid state, meaning more than 123% increase of the time-average charging rate over the surface irradiation mode. These results are coincided with that in Fig. 4e in which almost all the paraffin-graphene composite in inner-light-supply mode changes to liquid state while about two-thirds of the composite in surface irradiation mode are still in solid state. Remarkable increase of charging rate in inner-light-supply mode arises from the optimization of charging interface in spatial dimension as the side-glowing optical fiber localizes the light in the inner of the paraffin-graphene composite. It should be noted that the optical fibers will take up the volume of paraffin and thus decrease the energy storage density of the whole system. According to the volume ratio of the optical fiber to PCMs, the energy storage density will decrease by 6.3% here. This decrease could be greatly reduced with thinner fiber. Stability is a fatal factor in the practical application of phase change heat storage. We record the temperature evolution of the paraffin-graphene composite in inner-light-supply mode during 4 cycles of phase change. After continuous 4 cycles, the composite still has high charging rate and uniform temperature distribution (Fig. 4f and

Supplementary Movie 1), demonstrating the stability of device with this strategy.

## Numerical simulations of the solar-thermal conversion process

To exactly describe the faster charging rate of inner-light-supply mode, we analyzed the thermal energy utilization in these two modes. The realistic temperature distribution in these two modes was recorded by infrared camera. As shown in Fig. 5a, the sample in the surface irradiation mode has a temperature span of 90 °C with the maximum temperature of 160 °C (Supplementary Fig. 7). The temperature distribution in inner-light-supply mode is fairly uniform with the lowest temperature of 67 °C and the highest temperature of 80 °C (Fig. 5b). We then calculated the heat transport in these two modes. As shown in Fig. 5c, d, the heat loss in the form of radiation and convection in surface irradiation mode is significantly larger than that in inner-light-supply mode, and the resulted energy storage efficiency is 65.9% and 94.85%, respectively. We then used a finite element method to simulate the heat conduction process in these two modes and the temperature distribution is shown in Figs. 5e, f and Supplementary Fig. 8. It can be seen that the inner-light-supply mode changes the thermal transport path in spatial dimension. As shown in Fig. 5e, after charging for 2190 s, the highest temperature in the surface irradiation mode reaches up to 435 K and decreases to 300 K along the axial direction with a gradient of 135 K. At the same time, the highest temperature in the inner-light-supply mode is found near the optical fiber at 371 K and decreases to 315 K along the radial direction with a gradient of -56 K (Fig. 5f). This demonstrates that heat is localized in the inner part in inner-light-supply mode, which is more beneficial to thermal storage. Moreover, we tracked the phase state evolution process of paraffin-graphene composites in these two modes (Fig. 5g and Fig. 5h). In surface irradiation mode, the paraffin-graphene composite quickly melted from top side to the bottom side after charging in 1500 s. And continuing to prolong the charging time the melt speed of composite is slow and only 40% of the composite is melted to liquid after charging for 2190 s. While in inner-light-supply mode, the paraffin-graphene composite near the optical fiber area melts preferentially and all of the composite

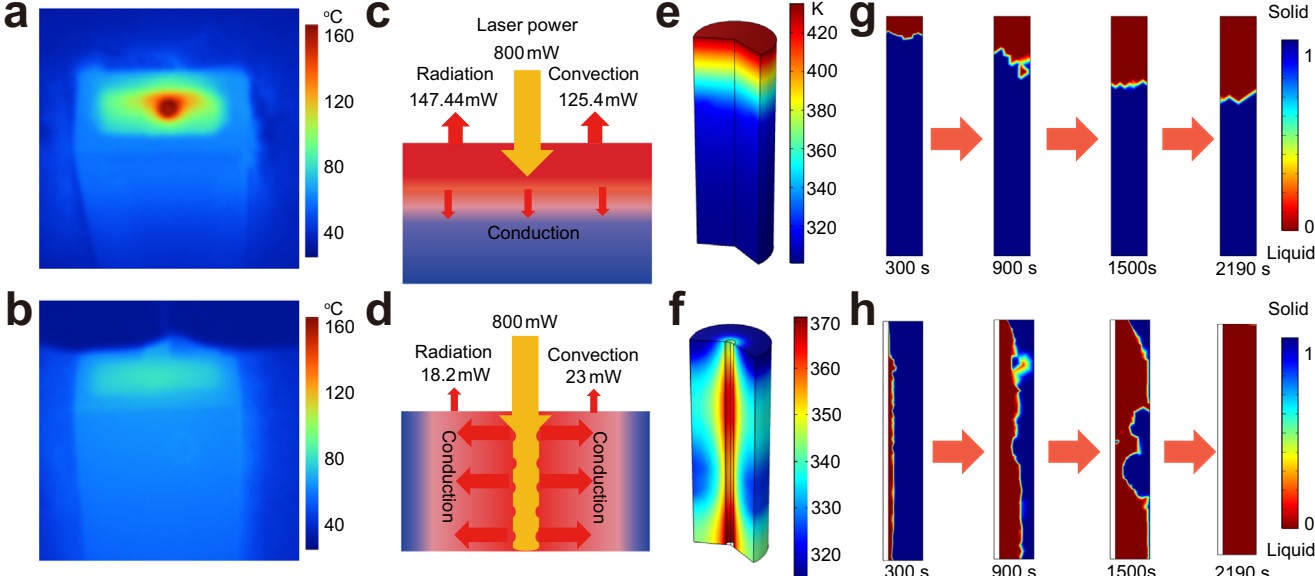

**Fig. 5 | Theoretical modeling of the charging of paraffin-graphene composites.** **a**, **b** IR thermal image of the top surface of paraffin-graphene composites in surface irradiation mode (**a**) and in inner-light-supply mode (**b**). **c**, **d** Schematic illustration of energy flows of paraffin-graphene composites in surface irradiation mode (**c**) and in inner-light-supply mode (**d**). **e**, **f** Simulated temperature distribution of paraffin-graphene composites in surface irradiation mode (**e**) and in inner-light-supply mode (**f**) after charging for 2190 s. **g**, **h** Simulated phase state distribution of paraffin-graphene composites in surface irradiation mode (**g**) and in inner-light-supply mode (**h**) after charging for 300 s, 900 s, 1500 s, and 2190 s. Zero stands for liquid state and one stands for solid state in the scale bar.

is melted to liquid after charging for 2190 s. The calculated temperature distribution and phase state distribution are close to the experimentally measured temperature distribution and phase state distribution (Fig. 4). We also studied the performance of inner-light-supply mode with different thermal conductivity of the PCM. As shown in Supplementary Fig. 9, the simplified parameter was adopted as assuming the thermal conductivity of the solid PCM and liquid PCM increased in the same proportion. In surface irradiation mode, when the thermal conductivity increases by 100 percent, the charging rate obviously increases (Supplementary Fig. 9a). When optical fiber is used, the charging rate can further increase (Supplementary Fig. 9b). It takes about 2190 s to melt all the composite when the thermal conductivity is increased by 270 percent in surface irradiation mode (Supplementary Fig. 9c), and decreases to 1485 s in inner-light-supply mode (Supplementary Fig. 9d). This indicates that the inner-light-supply mode can work excellent together with other strategies in which the thermal conductivity of PCMs is enhanced. Comparing Supplementary Fig. 9c and Fig. 5h, it takes the same time to melt all the composite. Thus we can conclude that the optical fiber shows similar effect in enhancing the charging rate to improving the thermal conductivity by 270 percent. In addition, it should be noted that as a simplified prototype, the operating distance of the optical fiber is not optimized here. The enhancement effect of the inner-light-supply mode in charging rate may be underestimated here.

## Field tests

To further demonstrate the practical feasibility of this inner-light-supply phase change system, we performed an outdoor experiment in one sunny autumn day (Aug. 24, 2022). Fresnel lens was used to focus sunlight and transmit it into the optical fiber (Fig. 6a, b). As a contrast, a surface irradiation mode phase change system (Fig. 6c) was measured in the same time. The solar flux and ambient temperature are shown in Fig. 6d. The time-sequential IR images in Fig. 6e show that the charging interface moves faster obviously than that of the surface irradiation

mode. After 65 min of illumination all the composite in the inner-light-supply mode is melted while about 40% of the composite in surface irradiation mode melts to liquid. At last, we collected 7.4 g of liquid in inner-light-supply mode and 3.2 g of liquid state composite in surface irradiation mode (Fig. 6f, g). Therefore, we can conclude that under the same solar illumination the average charging rate in surface irradiation mode is about 40% of that in the inner-light-supply mode.

For potential practical solar-thermal storage on large scale, we performed an outdoor experiment using a high-capacity (500 mL) container loaded with the paraffin-graphene composite (Fig. 7a). A lens with a diameter of 50 cm was used to condense the sunlight. Within this container, an array of seven side-glowing optical fibers was uniformly fixed in the paraffin-grapheme composite (Fig. 7b and Fig. 7c). The time-consequent IR images in Fig. 7d, e and Supplementary Fig. 10 show the evolution of temperature distribution in this bulk sample. All of paraffin-graphene composite became molten state after 80 min, indicating that this strategy can be adopted in large-scale solar thermal storage system.

In perspective, the inner-light-supply mode can avoid the overheating of the surface and reduce the heat loss from surface during the photo-thermal conversion process. The long-distance light conduction characteristic of optical fiber shortens the heat transfer distance and circumvent the quickly decayed heat diffusion in PCM, which enables the fast solar-thermal energy harvesting in large-scale STES. Thus inner-light-supply mode may be more valuable when it is used in a concentrate solar plant, in which silicon dioxide optical waveguide could be used to adapt the high temperature. What is more, the inner-light-supply strategy can be further optimized when combing with other engineering strategy. For example, the optical fiber can be coated with heat conducting tube. Thus the heat release of the thermal storage system can be enhanced.

## Discussion

In summary, we introduced optical waveguide into solar-thermal energy storage system to enhance the charging rate and solar-thermal

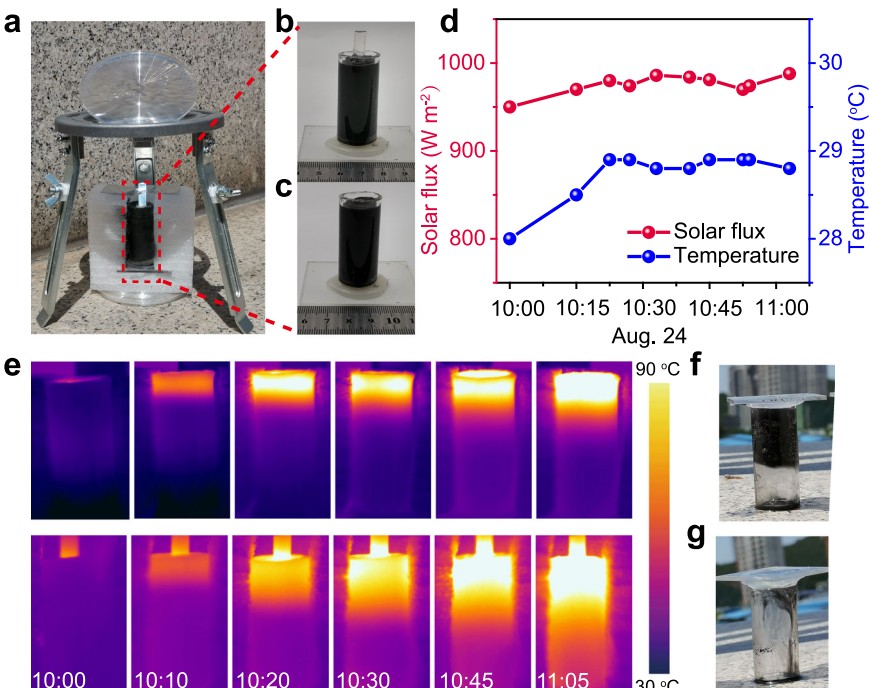

**Fig. 6 | Outdoor experiments with the inner-light-supply mode enhanced phase change thermal storage system. a** Experimental setup of outdoor experiments. **b, c** Photograph of the paraffin-graphene composites in the surface irradiation mode (**b**) and inner-light-supply mode (**c**). **d** Solar radiation and ambient temperature from 10:00–11:05. **e** Temperature evolution of paraffin-graphene composites in surface irradiation mode (top) and inner-light-supply mode (bottom) by prolonging the charging time from 10:00 AM to 11:05 AM. **f, g** Photographs of the paraffin-graphene composites of surface irradiation mode (**f**) and inner-light-supply mode (**g**) after charging for 65 min out of doors. In order to get a better view, we put the sample upside down.

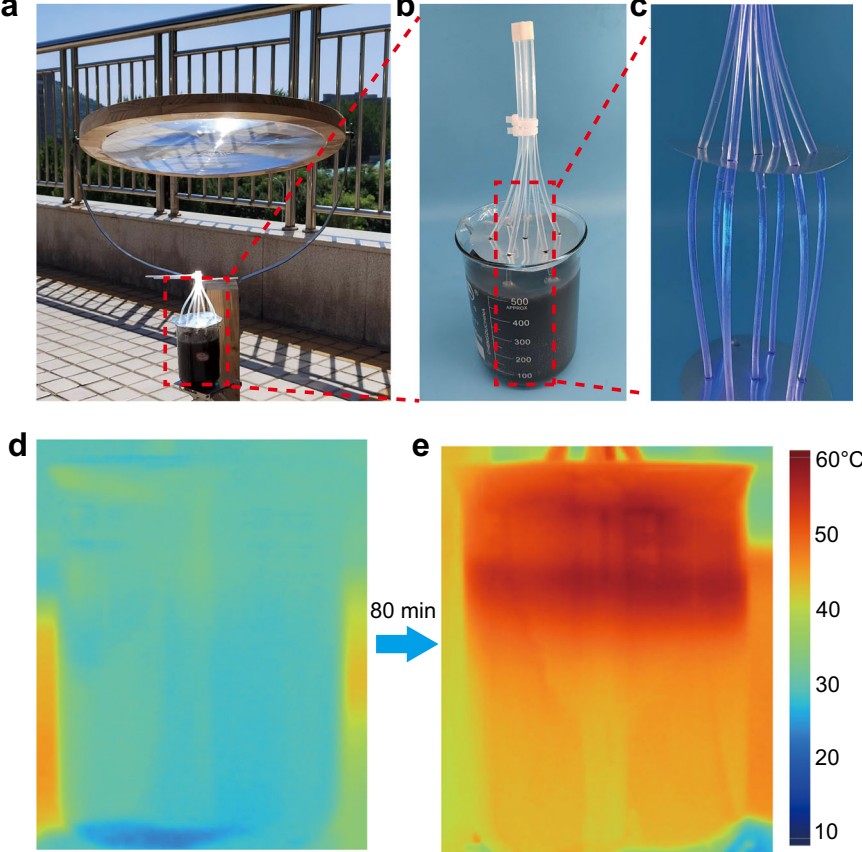

**Fig. 7 | Solar thermal charging of the large-scale inner-light-supply mode enhanced phase change thermal storage system. a** Outdoor experimental setup of large scale solar-thermal energy storage system. **b** Photograph of the paraffin-graphene composite bulk sample in inner-light-supply mode. **c** Photograph of side-glowing optical fiber array. **d**, **e** Temperature distribution of the sample at the beginning (**d**) and ending (**e**) of the experiment.

energy conversion efficiency. PMMA side-glowing optical fiber was prepared and was used to guide the incident light into the inner of the composite PCM. This optimization of solar-thermal charging interface avoided the overheating surface of the PCMs and reduced the convection and radiation heat loss greatly. Comparing to traditional surface irradiation mode, this inner-light-supply mode accelerated the charging rate by 123% and the solar thermal efficiency could up to 94.85%. Moreover, this inner-light-supply mode enhanced STES displayed excellent cycling stability and could work efficiently in large-scale outdoors. The work offers an avenue for high-performance PCMs in solar energy utilization and may extend the application of optical waveguide theory.

## Methods

### Materials

Acetone (99.5%) and n-hexane (99.5%) were obtained from Yantai Far East Fine Chemicals Co., Ltd (Yantai, China), Sodium dodecyl benzene sulfonate (SDBS, 88%) was purchased from Sinopharm Chemical Reagent Co., Ltd (Shanghai, China). Paraffin was supplied by Shanghai Maclin Biochemical Technology Co., Ltd (Shanghai, China). Graphene (98%) was purchased from Shenzhen Wenheng Technology Co., Ltd (Shenzhen, China). Ultrapure water with a resistivity of approximately 18.25 MΩ was used as fiber cleaning agent.

### Preparation of paraffin-graphene composite

Paraffin-graphene composite was prepared by dispersing few-layer graphene in the common commercially available organic paraffin with the aid of surfactant. Specifically, different amount of the mixture of graphene and SDBS with the ratio of 1:1 was dispersed into 40 g molten paraffin in a 100 mL glass beaker and then was ultrasonically treated at 60 W for 4 h to form a homogenous solution.

### Preparation of side-glowing optical fiber

The side-glowing optical fibers were prepared via solvent-etching method. In general, the etching solvent was prepared by mixing acetone and hexane with a volume ratio of 2:3. The cladding of the commercial polymer optical fibers (POF) was physically removed and the core of the POF (polymethyl methacrylate, PMMA) was immersed in the etching solvent for different times. Then the fiber core was rinsed twice with ultrapure water to remove the residual solvent.

### Characterizations

Scanning electron microscopy (SEM) images were obtained on a Hitachi Regulus-8100 Field Emission Scanning Electron Microscope. The diffuse reflectance of the solid piece of the PARAFFIN@graphene composite with different loading of graphene was measured using an ultraviolet-visible-near infrared (UV-vis-NIR) spectrophotometer (Hitachi UH4150, Japan) coupled with a φ150 mm integrating sphere. These samples were prepared to be solid slices with a thickness of 1 mm. DSC data were obtained with a differential scanning calorimeter (TA-DSC25-00834) with an Al 30 μL pan. All the sample weights are between 6 and 9 mg, and the temperature change rate is 5 °C under the protection of argon (Ar).

### Side glowing property measurements

The side glowing property of the optical fibers was recorded by the Digital Single Lens Reflex camera (Canon 90D camera) (Supplementary Fig. 11). Note that the afterglow images required the exposure time

of 1/8000 s and the ISO of 100. To avoid the interference of incident illumination to the side glowing properties of optical fiber, the optical fiber was fixed in a laser visor. The concentrated laser (800 mW) from an optical maser was directly shed on the end face of the fiber.

## Characterization of thermal energy storage

Thermal conductivities of paraffin and paraffin-graphene composite were measured using a steady-state method with homemade test equipment[59]. During the charging process, the obtained solid paraffin and paraffin-graphene composite were loaded into a quartz container (1 cm × 0.5 cm × 4 cm). The whole sample was thermally insulated by polymer foams except the top light incident surface and the front temperature monitoring side. In inner-light-supply mode, blue laser (450 nm, 800 mW) with a beam diameter of 1.5 mm was channeled through a side-glowing optical fiber with the diameter of 2 mm into the paraffin-graphene composite. In surface irradiation mode, the paraffin-graphene composite was optically charged by direct illumination of this blue laser. Temperature distribution of the paraffin-graphene composite was recorded with a thermal infrared camera (FLUKE Ti10 and FOTRIC 226) from the front surface and analyzed using Smart View 4.1 software.

## Field tests

In outdoor performance, the solid paraffin-graphene composite was loaded into a cylindrical container with a radius of 2 cm and a height of 5 cm or into a 500 mL beaker. In inner-light-supply mode, the sunlight in nature was condensed by a Fresnel lens with a diameter of 8 cm and 50 cm, respectively. The condensed light was transmitted to the inner of paraffin-graphene composite with a side-glowing optical fiber with a diameter of 6 cm or with the side-glowing optical fiber array which is composed of seven optical fibers with a diameter of 4 cm. In surface irradiation mode, the paraffin-graphene composite was optically charged by the condensed light. Temperature distribution of the paraffin-graphene composite was recorded with a thermal infrared camera (FLUKE Ti10 and FOTRIC 226) from the front surface and analyzed using Smart View 4.1 software.

## Reporting summary

Further information on research design is available in the Nature Portfolio Reporting Summary linked to this article.

# Data availability

The datasets generated during the current study are available from the corresponding author on request. Source data are provided with this paper.

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

## Acknowledgements

This work was financially supported by National Natural Science Foundation of China (Grant No. 51732007), Taishan Scholars Program of Shandong Province (Grant No. tsqn201812085), The Project of "20 Items of University" of Jinan (Grant No. 202228078), Doctoral foundation of University of Jinan (Grant No. 160100405), National Natural Science Foundation of China (Grant No. 51903102), Natural Science Foundation of Shandong (Grant No. ZR2022MB142). The manuscript was written through contributions of all authors. All authors have given approval to the final version of the manuscript.

## Author contributions

G.X. and H.L. conceived, designed, and supervised this study. Y.Z., J.T., J.C., M.D., W.Z., and X.X. performed the experiments and discussed the data. Y.Z. conducted the simulations. X.C. and Y.H.Z. performed the side glowing property measurements. G.X., H.L., and Y.Z. analyzed the results with all authors and wrote the manuscript.

## Competing interests

The authors declare no competing interests.
