## [Peer Review File · Nature Communications]

Accelerating the solar-thermal energy storage via inner-light supplying with optical waveguideReviewers' comments:

Reviewer #1 (Remarks to the Author):

This article introduces an interesting work. The authors use scattering optical fiber to guide the focused sunlight or laser into the black-colored phase change materials (PCMs), effectively reducing the convection heat loss and radiative heat loss due to high surface temperature. This conclusion has been experimentally verified in a simple graphene doped paraffin PCMs, and has also been supported theoretically by finite element simulations. Considering that this work pays more attention to the photo-thermal charging method than the material itself, and its application purpose is very clear, there are some places that need to be supplemented and explained to highlight the value of this work for phase change materials in the field of photo-thermal energy storage.

1 Photo-thermal conversion is a process of light response, which is not only related to the depth of light transmission, but also related to the light receiving area. Optical fiber is a typical device with large aspect ratio, and its constructed path is of great help to deeper energy storage tanks. Similar works of photo-thermal phase change energy storage that have been reported commonly use sheet samples for photo-thermal measurement, many of which reach conversion efficiency more than 90% (ACS Sustain. Chem. Eng., 7 (2019), pp. 17523-17534, Appl. Energy, 237 (2019), pp. 83-90, Energy Storage Mater., 20 (2019), pp. 401-409, J. Mater. Chem. A, 7 (2019), pp. 14319-14327,), almost no inferior to this work. Therefore, this paper has insufficient explanation on the effectiveness and application scope of its strategies enhancing solar thermal efficiency with optical waveguide. If the applied scene is large in surface area but not deep enough, how can this strategy work? The author had better study the enhancement effect and application value of this strategy in phase change materials with different shapes, such as thinner sheets and super long columns.

2 Photo-thermal energy storage involves heat conduction process, because radiative sunlight is generally absorbed by a thin layer of absorbing material (e.g. graphene) on the surface or interface of the composite PCMs, and then transfers heat to the unexposed area. Therefore, high thermal conductivity is helpful for temperature diffusion, thus reducing the environmental heat loss caused by radiation and convection during charging, and avoiding local overheating problem under focused sunlight. In this work, the optical waveguide can reduce the average heat transfer path during the charging process, thereby reducing the demand for thermal conductivity. The disadvantages of low thermal conductivity are introduced in the introduction but the author only measured the low thermal conductivity of the material, and did not further explore the influence of thermal conductivity on the strategy effect. More exploration will help to strengthen the persuasiveness of this article.

3 The longitudinal light receiving area can be increased by conducting light through scattering optical fiber. Since the scattering intensity is basically caused by random etching damages, it can be considered that the average scattering rate of each cross section of the fiber is approximate. Then the scattered light will have intensity distribution with depth along the fiber, meaning the intensity will be lower near the bottom. For an optical fiber with certain parameters, there will be an applicable depth limit, which is critical for practical applications. It is recommended to supplement relevant research.

4 For large-scale heat storage applications, rapid charging with high conversion efficiency is critical for the application of PCMs, but the heat release of phase change heat storage systems is also worthy of study. The insertion of optical fiber and the low thermal conductivity make the PCMs absorb sunlight more efficiently, but make the utilization of the stored heat energy more complex and difficult. Will this problem become an unsolvable defect of the strategy?

5 The energy storage density is an important parameter for the energy storage system, not just the phase change enthalpy of materials. In this paper, the author added a small amount of graphene, which has negligible impact on the overall energy storage density. However, the addition of enough optical fibers has a significant impact on the energy storage density of the system, which requires detailed quantitative research.

6 In this paper, the solar absorption characteristics of the materials were characterized by ultraviolet visible near-infrared absorption spectroscopy, and the effects of the amount of graphene added were compared. But there were obvious mistakes. The author only uses the reflectivity to obtain the absorptivity without considering the transmissivity, otherwise there will be no obvious error that the solar light absorptivity of pure paraffin exceeds 70% in Fig.2b. (paraffin has almost no absorption in the visible light and near IR region, see Nat Commun 8, 1478 (2017), Energy Stor. 13 (2018) 88–95). Relevant tests need to be redesigned and relevant data needs to be re-measured.

7 There are some inaccurate expressions such as “the most comment strategy” in Line 40, Page 2, the singular and plural forms of verbs in Line 187, Page 10 and so on.

Although this work is very interesting, more in-depth and more complete research is needed in many aspects (Q1-3, Q5). Moreover, there are some basic common-sense mistakes and incorrect results(Q6). Therefore, publication is not recommended., I think this article should be further improved and not proper for publication in Nature Communications.

Reviewer #2 (Remarks to the Author):

The authors present a novel method for directly charging a phase-change material (PCM) thermal storage using incoming solar radiation. The concept is novel, and opens up interesting methods for charging PCMs using solar radiation. However, there are some issues with the manuscript that must be addressed prior to publication.

1) The authors should add a figure similar to Figure 1 that shows the alternative ‘surface radiation only’ configuration. It seems like in Figure 5 the radiation is being applied by a laser, but sunlight does not come to the earth in a concentrated laser, but would instead be spread out over the entire surface of the PCM. This seems like a better baseline. Or is the system meant to be connected to a concentrated solar power plant? I would think a concentrated solar collector would be designed to go to a higher temperature than the paraffin selected ($T_t \sim 42$ C).

2) Also related to the baseline for comparison, there are many enhancements that increase conductivity of a PCM by 50-100x (e.g., Mills et al, 2006; Ji et al, 2014). However, these require more graphite material, and therefore displaces more of the paraffin. But the optical fibers also displace the paraffin. I suggest the authors add another comparison of a high conductivity PCM composite. What conductivity of this composite is needed to reach the equivalent performance of the optical fiber design? I’m thinking specifically of Figure 4b.

3) The authors should add the definition of efficiency. I presume it is heat collected in the storage vs. the incoming solar radiation. But that is what is used for solar thermal collectors operating at steady state. But the experiment presented here is a transient process, with the PCM surface increasing in temperature with time. Is the efficiency calculated based on cumulative heat collected over the full charging time vs. incident solar radiation over that same time?

4) For the efficiency and charging rate, the power going in can be measured by monitoring the electric power consumption, but the thermal energy in the PCM is more difficult to measure. How was it measured? Were the IR images used to quantify the enthalpy of the PCM? Were the embedded thermocouples? Or was it by weighing the liquid at the end of the test? If done by temperature, there is some PCM that is at a temperature within the phase-change region, and it is difficult to know how much of the PCM is liquid or solid, especially with uncertainty in the temperature measurement. The authors should explain how they took the measured data and converted that into high-level metrics like charging rate, solar thermal efficiency, or fraction of PCM that is melted.

5) Related to this calculation, the DSC was used at 5 degC/min ramp rate, which always leads to a wider phase-change region than the actual material. This DSC-created enthalpy-temperature curve cannot be used for the analysis because the actual material has a much smaller temperature range from fully liquid to fully solid. In my experience a 5 degC/min ramp rate can make the peak 2-3x wider than the actual material.

6) It is unclear what the PCM can be used for, and how it is discharged. The optical fibers will melt the PCM to charge it, but how will it be discharged? If it is with a pumped fluid within embedded

tubes, then why not just heat the fluid with a solar thermal collector and use that fluid to charge/melt the PCM? There are likely advantages, but this should be discussed to show how the proposed concept is beneficial compared to a very basic and common alternative. Shining a laser at the center of a PCM is not a common alternative method for PCM storage for solar thermal collectors.

7) The authors state, "...and paraffin-graphene composites with paraffin loadings ranging from 0.01 wt% to 0.06 wt%." Should this be graphene loadings instead of paraffin loadings?

8) In Figure 4, are (c) and (d) captions switched? I don't understand the results if they are not. (c) looks like the inner light mode based on the Figure 4 (b). Please check all other figures and captions, including figure 4d, figure 6 f, g, etc.

9) In Figure 1, why is the collection surface called a condenser? When I see "condenser" I think of a heat exchanger that condenses a fluid (e.g., a refrigerant)

10) Please proofread the manuscript. There are multiple instances of using "melt" when it should be "melted." For example, "...all the composite in the inner-supply mode is melt while only about 40% of the solid composite in surface irradiation mode melt to liquid." The first melt should be melted, while the second should be melts. Also, why does it say "40% of the solid composite" melts to liquid. Can't this just say 40% of the composite? Is there already liquid, and therefore it is only 40% of the PCM that is originally solid? The very next sentence says, "we collected 7.4 g composite of liquid state..." Isn't this just "we collected 7.4 g of liquid?" Or is it 7.4 g of liquid + graphite, and therefore it needs to specify composite? These sentences are confusing. There are also typos, like the use of 'obvious' instead of 'obviously', 'wase' instead of 'was' in the methods section. That same sentence also says, "physical removed" but should be "physically removed." Later 'lase' should be 'laser.' Please check the entire article.

References:

Ji, H., D.P. Sellan, M.T. Pettes, X. Kong, J. Ji, L. Shi, R.S. Ruoff. Enhanced thermal conductivity of phase change materials with ultrathin-graphite foams for thermal energy storage. *Energy & Environmental Science*. 7(3) (2014) 1185-92.

Mills, A., M. Farid, J.R. Selmán, S. Al-Hallaj. Thermal conductivity enhancement of phase change materials using a graphite matrix. *Appl. Therm. Engr.* 26(14) (2006) 1652-61.

Reviewer #1: This article introduces an interesting work. The authors use scattering optical fiber to guide the focused sunlight or laser into the black-colored phase change materials (PCMs), effectively reducing the convection heat loss and radiative heat loss due to high surface temperature. This conclusion has been experimentally verified in a simple graphene doped paraffin PCMs, and has also been supported theoretically by finite element simulations. Considering that this work pays more attention to the photo-thermal charging method than the material itself, and its application purpose is very clear, there are some places that need to be supplemented and explained to highlight the value of this work for phase change materials in the field of photo-thermal energy storage.

Response: Thanks very much for your professional advice. We really appreciate your help to improve our manuscript. The manuscript has been revised according to your constructive suggestions and the revisions were highlighted with blue color. Your specific comments have been responded point by point as follows.

Comment 1. Photo-thermal conversion is a process of light response, which is not only related to the depth of light transmission, but also related to the light receiving area. Optical fiber is a typical device with large aspect ratio, and its constructed path is of great help to deeper energy storage tanks. Similar works of photo-thermal phase change energy storage that have been reported commonly use sheet samples for photo-thermal measurement, many of which reach conversion efficiency more than 90%(ACS Sustain. Chem. Eng., 7 (2019), pp. 17523-17534, Appl. Energy, 237 (2019), pp. 83-90, Energy Storage Mater., 20 (2019), pp. 401-409, J. Mater. Chem. A, 7 (2019), pp. 14319-14327.), almost no inferior to this work. Therefore, this paper has insufficient explanation on the effectiveness and application scope of its strategies enhancing solar thermal efficiency with optical waveguide. If the applied scene is large in surface area but not deep enough, how can this strategy work? The author had better study the enhancement effect and application value of this strategy in phase change materials with different shapes, such as thinner sheets and super long columns.

Response: Thanks for your suggestions.

1) Heat conduction obeys Fourier's law as $Q = -\lambda \frac{T_1 - T_2}{\delta} A$, where δ is distance along the temperature gradient. It is obvious that the thickness of sample plays an important role in conducting heat. So charging the PCM with the form of thin film is an effect strategy to enhance the charging rate. For example, the reference [56] (J. Mater. Chem. A 2020, 8, 20970-20978) reported a roll-to-roll solar-thermal energy harvesting system to shorten heat-diffusion distance. The essence of introducing optical waveguide here is also to shorten heat-diffusion distance as light is transmitted into the inner of the bulk PCM with optical fiber. Thus its advantage could more exhibit in super long columns comparing to thinner sheets. In the revised manuscript, we have stressed that the applied scene is to accelerate the charging rate in **bulk** PCM.

2) Furthermore, the effect of thermal conductivity in inner-light-supply mode was studied with COMSOL simulation. It shows that the inner-light-supply mode can work excellent when the thermal conductivity of PCM is enhanced. Thus we demonstrate that this strategy is not to replace existing solar-thermal storage systems, and it can be adopted in existing STES systems in which the thermal conductivity of PCM is enhanced.

3) In addition, the solar-thermal conversion efficiency of the sheet sample is controversial. According to the references, the surface temperature of the thin film under one sun is about 70 °C. The corresponding surface heat loss in the form of radiation and convection can be calculated as $\varepsilon\sigma(T_0^4 - T_{amb}^4) + h(T_0 - T_{amb})$, where T_0 is the surface temperature of PCM, T_{amb} is the ambient temperature (assumed to be 30 °C), ε is the surface emissivity of PCM (assumed to be 1), σ is the Stefan-Boltzmann constant and h is the heat transfer coefficient (assumed to be 5 W m⁻² K⁻¹). The calculated heat loss is about 507 W m⁻² and thus the solar-thermal efficiency cannot over 50%.

In the revised manuscript, we make a clear sense of the application prospect of this work. The commended references were cited in the revised manuscript as [24], [38], [45], [46]. The manuscript was revised as follow.

...However, most PCMs own low thermal conductivity which restricts the thermal charging rate in bulk samples... (Abstract section)

...The thermal conductivity of most PCMs is generally lower than $1 \text{ W m}^{-1} \text{ K}^{-1}$, which severely impedes the efficient thermal transfer in bulk PCM^{13, 14}...

...We also studied the performance of inner-light-supply mode with different thermal conductivity of the PCM. As shown in Fig. S8, the simplified parameter was adopted as assuming the thermal conductivity of the solid PCM and liquid PCM increased in the same proportion. In surface irradiation mode, when the thermal conductivity increases by 100 percent, the charging rate obviously increases (Fig. S8a). When optical fiber is used, the charging rate can further increase (Fig. S7b). It takes about 2190 s to melt all the composite when the thermal conductivity is increased by 270 percent in surface irradiation mode (Fig. S7c), and decreases to 1485 s in inner-light-supply mode (Fig. S7d). This indicates that the inner-light-supply mode can work excellent together with other strategies in which the thermal conductivity of PCM is enhanced. Comparing Fig. S7c and Fig. 5g, it takes the same time to melt all the composite. Thus we can conclude that the optical fiber shows similar effect in enhancing the charging rate to improving the thermal conductivity by 270 percent. In addition, it should be noted that as a simplified prototype, the operating distance of the optical fiber is not optimized here. The enhancement effect of the inner-light-supply mode in charging rate may be underestimated here...

Fig. S8 Simulated phase state distribution of paraffin-graphene composites with different thermal conductivity. **a, b** When the thermal conductivity is set to 2 times of original PCM, the melting process in surface irradiation mode (**a**) and in inner-light-supply mode (**b**). **c, d** When the thermal conductivity is set to 3.7 times of original PCM, the melting process in surface irradiation mode (**c**) and in inner-light-supply mode (**d**). Zero stands for liquid state and one stands for solid state in the scale bar.

Comment 2. Photo-thermal energy storage involves heat conduction process, because radiative sunlight is generally absorbed by a thin layer of absorbing material (e.g. graphene) on the surface or interface of the composite PCMs, and then transfers heat to the unexposed area. Therefore, high thermal conductivity is helpful for temperature diffusion, thus reducing the environmental heat loss caused by radiation and convection during charging, and avoiding local overheating problem under focused sunlight. In this work, the optical waveguide can reduce the average heat transfer path during the charging process, thereby reducing the demand for thermal conductivity. The disadvantages of low thermal conductivity are introduced in the introduction but the author only measured the low thermal conductivity of the material, and did not further explore the influence of thermal conductivity on the strategy effect. More exploration will help to strengthen the persuasiveness of this article.

Response: Thanks for your suggestions. The influence of thermal conductivity on the inner-light-supply mode was discussed with COMSOL simulation in the revised manuscript. As shown in Figure 5 and Figure S7, we simulated the charging rate in the thermal storage system with different thermal conductivity and heating mode. The revision was also given in comment 1.

Comment 3. The longitudinal light receiving area can be increased by conducting light through scattering optical fiber. Since the scattering intensity is basically caused by random etching damages, it can be considered that the average scattering rate of each cross section of the fiber is approximate. Then the scattered light will have intensity distribution with depth along the fiber, meaning the intensity will be lower near the

bottom. For an optical fiber with certain parameters, there will be an applicable depth limit, which is critical for practical applications. It is recommended to supplement relevant research.

Response: As shown in Figure 3, Figure S4 and Figure S5, we have studied the side-glowing properties of optical fiber with numerical simulation and experiment. We have showed that the side scattering intensity of the fiber is affected by the pothole size on the surface of the optical fiber. The size and density of the pothole can be adjusted with the etching time with the organic solvent. We gave a proper parameter according to the size of our device. If the length of the fiber is longer, it only need to adjust the etching condition to decrease the side light intensity. Therefore, the light can be evenly distributed according to actual demand. In addition, we provide example of commercial side-glowing optical fiber with large-area. The problem of depth limit may be easily solved in practical application.

Fig. S11 The photograph of large-area commercial optical waveguide fiber cloth. The inset shows the side-glowing property of the fiber cloth.

Comment 4. For large-scale heat storage applications, rapid charging with high conversion efficiency is critical for the application of PCMs, but the heat release of phase change heat storage systems is also worthy of study. The insertion of optical fiber and the low thermal conductivity make the PCMs absorb sunlight more efficiently, but

make the utilization of the stored heat energy more complex and difficult. Will this problem become an unsolvable defect of the strategy?

Response: Thanks for your suggestions. As discussed above, the inner-light-supply mode can further improve the thermal charging rate when the thermal conductivity of the PCM is enhanced. In the revised manuscript, we have emphasized that this inner-light-supply strategy was not to replace existing solar-thermal storage systems, and it can be adopted in existing STES systems in which the thermal conductivity of PCM was enhanced. For example, the inner-light-supply strategy can be further optimized by coating the optical fiber with heat conducting tube to enhance the heat release. We have revised the manuscript as follow.

...In perspective, the inner-light-supply mode can avoid the overheating of the surface and reduce the heat loss from surface during the photo-thermal conversion process. The long-distance light conduction characteristic of optical fiber shortens the heat transfer distance and circumvent the quickly decayed heat diffusion in PCM, which enables the fast solar-thermal energy harvesting in large-scale STES. Thus inner-light-supply mode may be more valuable when it is used in a concentrate solar plant, in which silicon dioxide optical waveguide could be used to adapt the high temperature. **What is more, the inner-light-supply strategy can be further optimized when combing with other engineering strategy. For example, the optical fiber can be coated with heat conducting tube. Thus the heat release of the thermal storage system can be enhanced...**

Comment 5. The energy storage density is an important parameter for the energy storage system, not just the phase change enthalpy of materials. In this paper, the author added a small amount of graphene, which has negligible impact on the overall energy storage density. However, the addition of enough optical fibers has a significant impact on the energy storage density of the system, which requires detailed quantitative research.

Response: Thanks for your suggestions. The main purpose of this work is introducing this inner-light-supply strategy and demonstrating its feasibility. To simplify the

experiment, the optical fiber was commercial goods and the size was not optimized. Further optimization should be made according to practical application. In this case, the guiding significance of quantitative research of the energy storage density is less. In addition, the effect of adding optical fibers on the energy storage density could be easily calculated. The effect mainly comes from the increased volume after adding optical fibers. So the decreased energy storage density can be calculated as $V_1/(V_1+V_2)E_s$, where V_1 and V_2 is the volume of optical fibers and paraffin, respectively. We have revised the manuscript as follow.

...In addition, it should be noted that as a simplified prototype, the operating distance of the optical fiber was not optimized here. The enhancement effect of the inner-light-supply mode in charging rate may be underestimated here...

...It should be noted that the optical fibers have negative impact on the energy storage density (E_s) of the system. The decreased energy storage density can be calculated as $V_1/(V_1+V_2)E_s$, where V_1 and V_2 is the volume of optical fibers and paraffin, respectively. But this could be greatly reduced with thinner fiber...

Comment 6. In this paper, the solar absorption characteristics of the materials were characterized by ultraviolet visible near-infrared absorption spectroscopy, and the effects of the amount of graphene added were compared. But there were obvious mistakes. The author only uses the reflectivity to obtain the absorptivity without considering the transmissivity, otherwise there will be no obvious error that the solar light absorptivity of pure paraffin exceeds 70% in Fig.2b. (paraffin has almost no absorption in the visible light and near IR region, see Nat Commun 8, 1478 (2017), Energy Stor. Mater. 13 (2018) 88–95). Relevant tests need to be redesigned and relevant data needs to be re-measured.

Response: Thanks for your suggestions. We carefully studied the references [Nat Commun. 2017, 8, 1478] and [Energy Stor. Mater. 2018, 13, 88–95]. The corresponding solar absorption characteristics were shown as follow.

Editorial Note: Top Figure reproduced from Wang, Z., Tong, Z., Ye, Q. et al. Dynamic tuning of optical absorbers for accelerated solar-thermal energy storage. *Nat Commun* **8**, 1478 (2017). <https://doi.org/10.1038/s41467-017-01618-w>

Bottom figure on this page has been redacted to remove third-party material where no permission to publish could be obtained.

Figure from [Nat Commun. 2017, 8, 1478]. The absorption spectrum of the Fe₃O₄@graphene dispersed within chloroform.

Editorial note: Figure redacted

It can be seen that they all use the absorbance unit. The absorbance of PW in [Energy Stor. Mater. 2018, 13, 88–95] is **0.4** in the range from 500 to 1000 nm. When the thickness of the measured sample is large enough, this “0.4” will have significant effect on the light absorptivity. For example, it is darkness in the depths of the sea through the seawater is “transparent”.

In our work, the absorptivity was obtained with considering the transmissivity and reflectivity. Integrating sphere was used to obtain the reflectivity of the absorber. The transmissivity of the composite was also added in the revised manuscript. As shown in Fig. S2, the transmissivity of the absorber is nearly zero as the thickness of the paraffin is about 1 mm. Thus we gave the absolute value of absorptivity of paraffin film of 1 mm. We owned corresponding experience in measuring the absorptivity, such as Chem.

Eng. J. 2022, 429, 132089 and Chem. Eng. J. 2022, 429, 132183. We have revised the manuscript as follow.

...**b** Absorption spectra of paraffin-graphene composites with a thickness of 1 mm in the range of 300 to 2500 nm...(caption of Fig. 2b)

Fig. S2 Transmissivity of paraffin-graphene composites with a thickness of 1 mm in the range of 300 to 2500 nm with different loading of graphene. The unit of the loading is percent.

Comment 7. There are some inaccurate expressions such as “the most comment strategy” in Line 40, Page 2, the singular and plural forms of verbs in Line 187, Page 10 and so on.

Response: Thanks for your suggestions. We have rechecked the manuscript and the manuscript was revised as follow.

...The most **common** strategy...

...The **cyan line** denoting the phase change temperature of the paraffin-graphene composite (42°C) is used to tracked the position of the charging interface...

Comment 8. Although this work is very interesting, more in-depth and more complete research is needed in many aspects (Q1-3, Q5). Moreover, there are some basic common-sense mistakes and incorrect results(Q6). Therefore, publication is not recommended., I think this article should be further improved and not proper for publication in Nature Communications.

Response: Thanks for your suggestions. We have try our best to demonstrate the

feasibility and scope of application of this strategy. Your questions are also all solved. We hope this revised manuscript could meet your requirements.

Reviewer #2: The authors present a novel method for directly charging a phase-change material (PCM) thermal storage using incoming solar radiation. The concept is novel, and opens up interesting methods for charging PCMs using solar radiation. However, there are some issues with the manuscript that must be addressed prior to publication.

Response: Thank you very much for this positive evaluation on this work. We really appreciate your help to improve our manuscript. The manuscript has been revised according to your constructive suggestions and the revisions were highlighted with blue color. Your specific comments have been responded point by point as follows.

Comment 1. The authors should add a figure similar to Figure 1 that shows the alternative ‘surface radiation only’ configuration. It seems like in Figure 5 the radiation is being applied by a laser, but sunlight does not come to the earth in a concentrated laser, but would instead be spread out over the entire surface of the PCM. This seems like a better baseline. Or is the system meant to be connected to a concentrated solar power plant? I would think a concentrated solar collector would be designed to go to a higher temperature than the paraffin selected ($T_t \sim 42$ C).

Response: Thanks for your suggestions. We also discussed this problem several times. In our original intention, the inner-light-supply mode is more valuable when it is used in a concentrate solar plant. This photo-thermal charging method can avoid the overheating of surface and reduce the heat loss from surface. The long-distance light conduction characteristic of optical fiber shortens the heat transfer distance and circumvent the quickly decayed heat diffusion of the PCM, which enables the fast solar-thermal energy harvesting in large-scale STES. However, considering the laboratory conditions, we present this prototype in a low temperature heat storage system.

In the revised manuscript, we have a clearer clarify of the application background. We have emphasized that this inner-light-supply strategy was not to replace existing solar-thermal storage systems, and it can be adopted in existing STES systems in which the

thermal conductivity of PCM was enhanced. For example, the inner-light-supply strategy can be further optimized by coating the optical fiber with a heat conducting tube to enhance the heat release. We have revised the manuscript as follow.

Fig.1 Schematic diagram of the phase-change thermal storage system. a In traditional surface irradiation mode, additive such as graphene is used to enhance the light absorption and thermal conductivity of the PCM. Solar-thermal conversion process occurs at the surface of the PCM. **b** To further accelerate the thermal charging rate, inner-light-supply mode is achieved with optical fiber. The sunlight is focused by collecting lens and then transmits into the PCM with the side-glowing optical fiber after multiple scattering and refracting. The solar-thermal conversion interface is localized in the inner of the PCM, in which well-dispersed graphene converts light to heat and heat is stored in PCM accompanying phase change process.

...In perspective, the inner-light-supply mode can avoid the overheating of the surface and reduce the heat loss from surface during the photo-thermal conversion process. The long-distance light conduction characteristic of optical fiber shortens the

heat transfer distance and circumvent the quickly decayed heat diffusion in PCM, which enables the fast solar-thermal energy harvesting in large-scale STES. Thus inner-light-supply mode may be more valuable when it is used in a concentrate solar plant, in which silicon dioxide optical waveguide could be used to adapt the high temperature. What is more, the inner-light-supply strategy can be further optimized when combining with other engineering strategy. For example, the optical fiber can be coated with heat conducting tube. Thus the heat release of the thermal storage system can be enhanced...

Comment 2. Also related to the baseline for comparison, there are many enhancements that increase conductivity of a PCM by 50-100x (e.g., Mills et al, 2006; Ji et al, 2014). However, these require more graphite material, and therefore displace more of the paraffin. But the optical fibers also displace the paraffin. I suggest the authors add another comparison of a high conductivity PCM composite. What conductivity of this composite is needed to reach the equivalent performance of the optical fiber design? I'm thinking specifically of Figure 4b.

Response: Thanks for your suggestions. In the revised manuscript, the effect of thermal conductivity in inner-light-supply mode was studied with COMSOL simulation. It shows that the inner-light-supply mode can work excellent when the thermal conductivity of PCM is enhanced. Thus we demonstrate that this strategy is not to replace existing solar-thermal storage systems, and it can be adopted in existing STES systems in which the thermal conductivity of PCM is enhanced. We have emphasized this in the revised manuscript as follow. The commended references were cited in the revised manuscript. The manuscript was revised as follow.

...We also studied the performance of inner-light-supply mode with different thermal conductivity of the PCM. As shown in Fig. S8, the simplified parameter was adopted as assuming the thermal conductivity of the solid PCM and liquid PCM increase in the same proportion. In surface irradiation mode, when the thermal conductivity increases

by 100 percent, the charging rate obviously increases (Fig. S8a). When optical fiber is used, the charging rate can further increase (Fig. S8b). It takes about 2190 s to melt all the composite when the thermal conductivity increases by 270 percent in surface irradiation mode (Fig. S8c), and decrease to 1485 s in inner-light-supply mode (Fig. S8d). This indicates that the inner-light-supply mode can work excellent together with other strategies in which the thermal conductivity of PCM is enhanced. Comparing Fig. S8c and Fig. 5g, it takes the same time to melt all the composite. Thus we can conclude that the optical fiber shows similar effect in enhancing the charging rate to improving the thermal conductivity by 270 percent. In addition, it should be noted that as a simplified prototype, the operating distance of the optical fiber was not optimized here. The enhancement effect of the inner-light-supply mode in charging rate may be underestimated here...

Fig. S8 Simulated phase state distribution of paraffin-graphene composites with different thermal conductivity. a, b When the thermal conductivity is set to 2 times of original PCM, the melting process in surface irradiation mode (a) and in inner-light-supply mode (b). **c, d** When the thermal conductivity is set to 3.7 times of original PCM, the melting process in surface irradiation mode (c) and in inner-light-supply mode (d). Zero stands for liquid state and one stands for solid state in the scale bar.

Comment 3. The authors should add the definition of efficiency. I presume it is heat

collected in the storage vs. the incoming solar radiation. But that is what is used for solar thermal collectors operating at steady state. But the experiment presented here is a transient process, with the PCM surface increasing in temperature with time. Is the efficiency calculated based on cumulative heat collected over the full charging time vs. incident solar radiation over that same time?

Response: Thanks for your suggestions. There are two ways to calculate the solar-thermal conversion efficiency. One is

$$\eta = \frac{mh}{APt}$$

where m is the mass, h is the phase change enthalpy, A is the surface area of the sample, P is the optical power density and t is the time taken for the phase transition.

and the other is

$$\eta = 1 - \frac{Q_r + Q_c}{E_{in}}$$

where Q_r and Q_c is radiative and convective heat energy loss from the top surface of container, E_{in} is the solar-thermal energy.

These two methods are both based on steady state. As the time taken for the phase transition is difficult to confirmed, we are more inclined to adopting the second one. And the calculated process is given in supporting information (Supplementary Note 2: Solar-thermal conversion efficiency) as follow.

...In the solar-thermal storage system, the PCM is surrounded by the thermal insulating polystyrene foam except the top surface. In ideal conditions, the heat dissipation mainly exists at the exposed top surface. Other heat loss, such as increasing the temperature of the container or heating up the insulation foam, is negligible. So the efficiency can be described as:

$$\eta = 1 - \frac{Q_r + Q_c}{E_{in}} \quad (3)$$

$$Q_r = A\varepsilon\sigma(T_0^4 - T_{amb}^4) \quad (4)$$

$$Q_c = Ah(T_0 - T_{amb}) \quad (5)$$

where Q_r and Q_c is radiative and convective heat energy loss from the top surface of container, E_{in} is the solar-thermal energy, A is the surface area of the top surface, T_0 is the surface temperature of PCM, T_{amb} is the ambient temperature, ε is the surface emissivity of PCM (assumed to be 1), σ is the Stefan-Boltzmann constant and h is the heat transfer coefficient (assumed to be $10 \text{ W m}^{-2} \text{ K}^{-1}$).

In the indoor experiment, the power of incident light is 800 mW and the ambient temperature is 30°C . According to the temperature distribution of the PW-composite samples in Fig. 4b, we extracted the temperature profiles (Fig. S6). It can be calculated with the numerical integration algorithm that the convection heat loss in the inner-light-supply mode is 23 mW and that of the surface irradiation mode is 259 mW. Moreover, taking the thermal emittance of 1, the radiative heat loss of the composites under these two modes is 18 mW and 230 mW, respectively.

Fig. S6 Temperature distribution profiles extracted from the IR thermal images of the top surface of paraffin-graphene composites. **a**, The temperature of the spots at different distances to the light spot center of the surface irradiation mode. **b**, The temperature distribution along the length of the quartz cuvette of the inner-light-supply mode.

Comment 4. For the efficiency and charging rate, the power going in can be measured by monitoring the electric power consumption, but the thermal energy in the PCM is

more difficult to measure. How was it measured? Were the IR images used to quantify the enthalpy of the PCM? Were the embedded thermocouples? Or was it by weighing the liquid at the end of the test? If done by temperature, there is some PCM that is at a temperature within the phase-change region, and it is difficult to know how much of the PCM is liquid or solid, especially with uncertainty in the temperature measurement. The authors should explain how they took the measured data and converted that into high-level metrics like charging rate, solar thermal efficiency, or fraction of PCM that is melted.

Response: Thanks for your suggestion. The solar-thermal conversion efficiency is calculated according to the heat loss (comment 3). As shown in Figure 4b, 4c and 4d, the IR images were used to quantify the thermal charging rate according reference (Nat Commun.,2017, 8, 1478). We have explained the calculating process of charging rate in the supporting information as follow.

Supplementary Note 2: Thermal charging rate and Solar-thermal conversion efficiency

The charging rate is confirmed by the translational speed of the charging interface. As shown in Figure 4c and 4d, the lines of temperature distribution at different time can be extracted from the infrared photo (Figure 4b). Then a cyan line denoting the phase change temperature of the composite (42°C) is used to tracked the position of the charging interface. The points of where these two lines intersect represent the position of the charging interface. The translational speed of these points represents the charging rate.

Comment 5. Related to this calculation, the DSC was used at 5 degC/min ramp rate, which always leads to a wider phase-change region than the actual material. This DSC-created enthalpy-temperature curve cannot be used for the analysis because the actual material has a much smaller temperature range from fully liquid to fully solid. In my experience a 5 degC/min ramp rate can make the peak 2-3x wider than the actual material.

Response: Thanks for your suggestions. We agree with you point. The ramp rate will

affect the shape of the enthalpy-temperature curve, especially when the thermal conductivity of the sample is very low. Here, we used the DSC to confirm the appropriate content of graphene. For easy comparison with other references (Adv. Funct. Mater. 2013, 23, 4354-4360; ACS Nano 2012, 6, 10884-10892; Nat Commun. 2017, 8, 1478), we chose the ramp rate of 5 °C min⁻¹. Since the influence of the ramp rate on the enthalpy is consistent with all samples, we can confirm the appropriate content of graphene from these data. In the following work, we will pay special attention to this point when we use this calculated enthalpy from DSC test.

Comment 6. It is unclear what the PCM can be used for, and how it is discharged. The optical fibers will melt the PCM to charge it, but how will it be discharged? If it is with a pumped fluid within embedded tubes, then why not just heat the fluid with a solar thermal collector and use that fluid to charge/melt the PCM? There are likely advantages, but this should be discussed to show how the proposed concept is beneficial compared to a very basic and common alternative. Shining a laser at the center of a PCM is not a common alternative method for PCM storage for solar thermal collectors.

Response: Thanks for your suggestion. As discussed in comment 1, in our original intention, the inner-light-supply mode was more valuable when it was used in a concentrate solar plant. However, considering the laboratory conditions, we present this prototype in a low temperature heat storage system. In the revised manuscript, we have a clearer clarify of the application background. We have emphasized that this inner-light-supply strategy was not to replace existing solar-thermal storage systems, and it can be adopted in existing STES systems in which the thermal conductivity of PCM is enhanced. For example, the inner-light-supply strategy can be further optimized by coating the optical fiber with a heat conducting tube to enhance the heat release. We have revised the manuscript as follow.

In perspective, the inner-light-supply mode can avoid the overheating of the surface and reduce the heat loss from surface during the photo-thermal conversion process. The long-distance light conduction characteristic of optical fiber shortens the heat transfer distance and circumvent the quickly decayed heat diffusion, which enables the fast

solar-thermal energy harvesting in large-scale STES. Thus inner-light-supply mode may be more valuable when it was used in a concentrate solar plant, in which silicon dioxide optical waveguide should be used to adapt the high temperature. What is more, the inner-light-supply strategy can be further optimized when combining with other strategies. For example, the optical fiber can be coated with heat conducting tube. Thus the heat release of the thermal storage system can be enhanced...

Comment 7. The authors state, "...and paraffin-graphene composites with paraffin loadings ranging from 0.01 wt% to 0.06 wt%." Should this be graphene loadings instead of paraffin loadings?

Response: Thanks for your carefulness. We have revised this mistake.

Comment 8. In Figure 4, are (c) and (d) captions switched? I don't understand the results if they are not. (c) looks like the inner light mode based on the Figure 4 (b). Please check all other figures and captions, including figure 4d, figure 6 f, g, etc.

Response: Thanks for your suggestion. We checked the figure caption and revised figure 4b. There is no mistake in figure 4c, d and figure 6f, g.

1) In figure 4b, the top is the surface irradiation mode and the bottom is the inner-light-supply mode. Figure 4c and Figure 4d are the lines of temperature distribution at different time extracted from the infrared photo (Figure 4b). The cyan line denotes the phase change temperature of the composite (42°C). The point of where these two lines intersect represents the position of the charging interface. It can be seen that the position of the charging interface in Figure 4d moved from 2 cm (15th min) to 4 cm (55th min), which is corresponding to the inner-light-supply mode (bottom of figure 4b).

2) In order to get a better view, we put the container upside down in Figure 6f and Figure 6g. Figure 6f is the photograph of the rest of composite in surface irradiation mode after charging for 65 min outdoors, in which the composite was not all melted. Figure 6g is the photograph of the rest of composite in inner-light-supply mode, in which the composite was all melted. The manuscript has been revised as follow.

Fig. 4 Charging of paraffin-graphene composite. **a** Schematic of experimental setup for charging of paraffin-graphene composite. A blue laser with the wavelength of 450 nm is used to illuminate onto the incident face of the POF in inner-light-supply mode and directly illuminate the surface of paraffin-graphene composite in surface irradiation mode. **b** Temperature evolution of paraffin-graphene composite in surface irradiation mode (top) and inner-light-supply mode (bottom) by prolonging the charging time from 0 min to 75 min. **c, d** Temperature distribution profiles at different charging times in surface irradiation mode (**c**) and inner-light-supply mode (**d**). The cyan line denotes the phase change temperature of the composite (42°C). **e** Photographs of paraffin-graphene composite in surface irradiation mode (right) and inner-light-supply mode (left) after charging for 75 min. In order to get a better view, we put the sample upside down. **f** IR thermal image of the paraffin-graphene composite in inner-light-supply mode during four phase transition cycles. During each cycle, we displayed the charging and discharging process.

Comment 9. In Figure 1, why is the collection surface called a condenser? When I see “condenser” I think of a heat exchanger that condenses a fluid (e.g., a refrigerant)

Response: Thanks for your suggestion. In heat transfer theory, condenser is usually a

heat exchanger that condenses a fluid. In optics, condenser is collecting lens. For easier reading, we have revised the “condenser” as “Fresnel lens”.

Comment 10. Please proofread the manuscript. There are multiple instances of using “melt” when it should be “melted.” For example, “...all the composite in the inner-supply mode is melt while only about 40% of the solid composite in surface irradiation mode melt to liquid.” The first melt should be melted, while the second should be melts. Also, why does it say “40% of the solid composite” melts to liquid. Can’t this just say 40% of the composite? Is there already liquid, and therefore it is only 40% of the PCM that is originally solid? The very next sentence says, “we collected 7.4 g composite of liquid state...” Isn’t this just “we collected 7.4 g of liquid?” Or is it 7.4 g of liquid + graphite, and therefore it needs to specify composite? These sentences are confusing. There are also typos, like the use of ‘obvious’ instead of ‘obviously’, ‘wase’ instead of ‘was’ in the methods section. That same sentence also says, “physical removed” but should be “physically removed.” Later ‘lase’ should be ‘laser.’ Please check the entire article.

References:

Ji, H., D.P. Sellan, M.T. Pettes, X. Kong, J. Ji, L. Shi, R.S. Ruoff. Enhanced thermal conductivity of phase change materials with ultrathin-graphite foams for thermal energy storage. *Energy & Environmental Science*. 7(3) (2014) 1185-92.

Mills, A., M. Farid, J.R. Selman, S. Al-Hallaj. Thermal conductivity enhancement of phase change materials using a graphite matrix. *Appl. Therm. Engr.* 26(14) (2006) 1652-61.

Response: Thanks for your carefulness. we have carefully checked the manuscript and revised the corresponding mistakes. The references were also cited in the revised manuscript as [47] and [31]. Some revisions are listed as follow.

...all the composite in the inner-light-supply mode is **melted** while about 40% of the composite in surface irradiation mode **melts** to liquid.

...the intensity of lateral light **obviously** increases

...The cladding of the commercial polymer optical fibers (POF) **was physically**

removed

REVIEWERS' COMMENTS

Reviewer #2 (Remarks to the Author):

The authors have addressed the comments adequately. My main hesitation was that the authors are demonstrating a novel phenomena, but did not clearly explain how it could be used / how it improves performance. It is clearer now that the authors have shown improved efficiency, and they have put it into better context by explaining more clearly how it would be used in an actual system.

Also, re: Figure 4c,d. These make sense. I think I was just reading the two modes backwards before.

I've summarized my thoughts at the bottom (comment 8), which is where reviewer 1 provided their high-level conclusions.

Reviewer 1:

Comment 1 - Inadequate explanation of effectiveness of approach and application (different form factor)

Response was adequate. There could be more added about the collector surface temperature being important in solar thermal collector efficiency, and how the inner-light supply mode reduces this surface temperature. But after reading further, this was covered adequately in response to comment 4.

Comment 2 – More on how inner-light supply mode could reduce surface temperature without high k.

Response to comment 4 really gets at what the reviewer was asking about, I think.

Comment 3 – asked about how far this fiber approach can penetrate into a PCM.

Authors did discuss this, but did not respond to the specific comment about there being "...an applicable depth limit..." I think the reviewer was asking, "how deep does this approach work?" It seems like this was adequately discussed in the response, but there could be more added about this in the manuscript. I didn't see the word "deep" or "depth" used anywhere in the manuscript. There's a mention of "operating distance" but it doesn't get at the question the reviewer was asking, as far as I can tell.

Comment 4 – inner light supply mode helps with charging, but what about discharging?

Adequate response here. Basically this is not the focus of the study, and improving charging efficiency with inner light supply mode can still be useful, regardless of how the system is designed to handle discharge.

Comment 5 – comment on impact of fibers on energy density

The authors addressed this adequately, but they put this in the SI. I would suggest mentioning it in the main body of the paper. But not required for acceptance. It'd also be nice if the authors actually calculated this, rather than just giving the reader the equation for them to calculate it. Even if it heavily impacts energy density here, that is okay, but it would give people a sense of where things stand w.r.t. energy density.

Comment 6 – comment on reflectivity / absorptivity / transmissivity

The response seems adequate to me, but I will be honest that I suspect reviewer 1 has more expertise in this area than I do.

Comment 7 – minor editorial.

Authors corrected this.

Comment 8 – Summary of above comments, and noting that comments Q1-Q3 and Q5 and Q6 make the paper unacceptable.

I believe responses to Q1-Q2 are adequate, and I think adding some more discussion on Q3 (describe impact of depth on performance in the manuscript, at least qualitatively) and Q5 (add the part about energy density to the manuscript, not the SI) would address these. However, I cannot confidently say whether comment Q6 has been adequately addressed or not. I don't see any major flaws, and I also don't think this impacts heavily the conclusions drawn by the authors. But I do not know enough about transmissivity of solids to comment further.

I agree with reviewer 1 that the authors glossed over some things, but I think the responses and edits addressed these. I also agree with reviewer 1 that the authors could have done a better job tying this to an application and explaining more clearly the benefits of this inner-light supply mode. But to me it still seems like an interesting study worthy of being published so others can potentially build upon (or even criticize) what was done by the authors.

Reviewer #2: The authors have addressed the comments adequately. My main hesitation was that the authors were demonstrating a novel phenomenon, but did not clearly explain how it could be used / how it improves performance. It is clearer now that the authors have shown improved efficiency, and they have put it into better context by explaining more clearly how it would be used in an actual system. Also, re: Figure 4c, d. These make sense. I think I was just reading the two modes backwards before. I've summarized my thoughts at the bottom (comment 8), which is where reviewer 1 provided their high-level conclusions.

Reviewer 1:

Comment 1 - Inadequate explanation of effectiveness of approach and application (different form factor)

Response was adequate. There could be more added about the collector surface temperature being important in solar thermal collector efficiency, and how the inner-light supply mode reduces this surface temperature. But after reading further, this was covered adequately in response to comment 4.

Comment 2 – More on how inner-light supply mode could reduce surface temperature without high k.

Response to comment 4 really gets at what the reviewer was asking about, I think.

Comment 3 – asked about how far this fiber approach can penetrate into a PCM.

Authors did discuss this, but did not respond to the specific comment about there being “...an applicable depth limit...” I think the reviewer was asking, “how deep does this approach work?” It seems like this was adequately discussed in the response, but there could be more added about this in the manuscript. I didn't see the word “deep” or “depth” used anywhere in the manuscript. There's a mention of “operating distance” but it doesn't get at the question the reviewer was asking, as far as I can tell.

Comment 4 – inner light supply mode helps with charging, but what about discharging? Adequate response here. Basically this is not the focus of the study, and improving charging efficiency with inner light supply mode can still be useful, regardless of how the system is designed to handle discharge.

Comment 5 – comment on impact of fibers on energy density

The authors addressed this adequately, but they put this in the SI. I would suggest mentioning it in the main body of the paper. But not required for acceptance. It'd also be nice if the authors actually calculated this, rather than just giving the reader the equation for them to calculate it. Even if it heavily impacts energy density here, that is okay, but it would give people a sense of where things stand w.r.t. energy density.

Comment 6 – comment on reflectivity / absorptivity / transmissivity

The response seems adequate to me, but I will be honest that I suspect reviewer 1 has more expertise in this area than I do.

Comment 7 – minor editorial.

Authors corrected this.

Comment 8 – Summary of above comments, and noting that comments Q1-Q3 and Q5 and Q6 make the paper unacceptable.

I believe responses to Q1-Q2 are adequate, and I think adding some more discussion on Q3 (describe impact of depth on performance in the manuscript, at least qualitatively) and Q5 (add the part about energy density to the manuscript, not the SI) would address these. However, I cannot confidently say whether comment Q6 has been adequately addressed or not. I don't see any major flaws, and I also don't think this impacts heavily the conclusions drawn by the authors. But I do not know enough about transmissivity of solids to comment further.

I agree with reviewer 1 that the authors glossed over some things, but I think the responses and edits addressed these. I also agree with reviewer 1 that the authors could have done a better job tying this to an application and explaining more clearly the benefits of this inner-light supply mode. But to me it still seems like an interesting study worthy of being published so others can potentially build upon (or even criticize) what was done by the authors.

Response: We sincerely thank the reviewer for taking time to review our work and helping to improve the manuscript. We have revised the manuscript according to Q3 (describe impact of depth on performance in the manuscript, at least qualitatively) and Q5 (add the part about energy density to the manuscript, not the SI) as follows. We also further updated the light absorption data according to Q6 to avoid the controversy.

1) Comment 3 – asked about how far this fiber approach can penetrate into a PCM.

Yes, we also think the reviewer was asking “what is the depth limit when the side-glowing fiber emits uniformly”. In the manuscript, we studied the variables that affect the side-glowing properties with numerical simulation and experiment. In the first Response, we only discussed the solution. Here, to clearly solve this problem, we added the discuss of “depth limit” into the revised manuscript as follow.

...It should be noted that the side glowing properties is also affected by length and diameter of the fiber. The optimized etching time is got at present length of the fiber. If the length of the fiber is various, the etching condition or etching method, such as laser corrosion, should be adjusted. As shown in Supplementary Fig. 5, commercial side-glowing optical fiber with large-area can be easily got. Thus the depth limit of uniform side-glowing fiber should be considered and may be easily solved in practical application.

2) Comment 5 – comment on impact of fibers on energy density.

We put the discussion of energy storage density into the main body of the revised manuscript. We also calculated the effect of the energy storage density after adding optical fiber. Since paraffin can be easily replaced by other phase change materials in our system, we give the decreased ratio of the energy density.

...It should be noted that the optical fibers will take up the volume of paraffin and thus decrease the energy storage density of the whole system. According to the volume ratio of the optical fiber to PCMs, the energy storage density will decrease by 6.3% here. This decrease could be greatly reduced with thinner fiber...

3) Comment 6 – comment on reflectivity / absorptivity / transmissivity.

In the previous response, we want to say “all the light except the reflected light will be absorbed by the composite”. This is maybe controversy. So we updated the light absorption ability of the composite with reflectivity and transmittance data which is directly got from the ultraviolet-visible-near infrared spectrophotometer.

...To quantitatively characterize the light absorption characteristic of paraffin-graphene composite, we measured the **transmittance and reflectivity spectra** ranging from 300 to 2500 nm. As shown in Fig. 2b and Supplementary Fig. 2, the absorption of pure paraffin is relatively weak **for its high transmittance and reflectivity** and can be obviously enhanced with a small amount of graphene. **The reflectivity is lower than 10% when the loading is over 0.02 wt%, indicating that over 90% incident light will be absorbed by the composite. Meanwhile, the transmittance is above 10% with the loading of 0.02 wt%, meaning light can transport over 1 mm in this composite.**

Fig. 2 Characterization of paraffin-graphene composite. **a** SEM image of the surface morphology of paraffin-graphene composite. **b** **Transmittance spectra of paraffin-graphene composites with a thickness of 1 mm in the range of 300 to 2500 nm spectra.** **c** DSC curves of pure paraffin and paraffin-graphene composites with different loading of graphene. **d** Comparison of fusion phase-change enthalpy, melting and solidification temperatures of pure paraffin and paraffin-

graphene composites.

Supplementary Fig. 2 **Reflectivity spectra** of paraffin-graphene composites with a thickness of 1 mm in the range of 300 to 2500 nm with different loading of graphene.